# Subcritical Water Extracts from *Agaricus blazei* Murrill’s Mycelium Inhibit the Expression of Immune Checkpoint Molecules and Axl Receptor

**DOI:** 10.3390/jof7080590

**Published:** 2021-07-23

**Authors:** Taro Yasuma, Masaaki Toda, Hajime Kobori, Naoto Tada, Corina N. D’Alessandro-Gabazza, Esteban C. Gabazza

**Affiliations:** 1Department of Immunology, Mie University Faculty and Graduate School of Medicine, Tsu 514-8507, Mie, Japan; t-yasuma0630@clin.medic.mie-u.ac.jp (T.Y.); t-masa@doc.medic.mie-u.ac.jp (M.T.); dalessac@clin.medic.mie-u.ac.jp (C.N.D.-G.); 2Iwade Research Institute of Mycology Co., Ltd., Tsu 514-0012, Mie, Japan; kobori@iwade101.com (H.K.); tada@iwade101.com (N.T.)

**Keywords:** mushroom, immune checkpoints, Axl receptor, lung cancer, dendritic cells, immune response

## Abstract

*Agaricus blazei* Murrill or Himematsutake is an edible and medicinal mushroom. *Agaricus blazei* Murrill’s fruiting body extracts have anticancer properties, although the mechanism is unknown. Basic or organic solvents, which are hazardous for human health, are generally used to prepare *Agaricus blazei* Murrill’s extracts. The inhibition of immune checkpoint molecules and Axl receptor is an effective therapy in cancer. This study assessed whether subcritical water extracts of the *Agaricus blazei* Murrill’s fruiting body or mycelium affect the expression of Axl and immune checkpoint molecules in lung cancer cells. We used A549 cells and mouse bone marrow-derived dendritic cells in the experiments. We prepared subcritical water extracts from the *Agaricus blazei* Murrill’s fruiting body or mycelium. The subcritical water extracts from the *Agaricus blazei* Murrill’s fruiting body or mycelium significantly inhibited the expression of immune checkpoint molecules and Axl compared to saline-treated cells. Additionally, the hot water extract, subcritical water extract, and the hot water extraction residue subcritical water extract from the *Agaricus blazei* Murrill’s mycelium significantly enhanced the expression of maturation markers in dendritic cells. These observations suggest that the subcritical water extract from *Agaricus blazei* Murrill’s mycelium is a promising therapeutic tool for stimulating the immune response in cancer.

## 1. Introduction

Mushrooms, members of the fungi kingdom, have been used in oriental medicine for therapeutic purposes since ancient times [1,2]. Mushrooms have been used empirically to treat inflammatory afflictions, infections, diabetic mellitus, cardiovascular diseases, and malignant tumors [3]. There is a huge number (1.5–5 million) of fungal species on earth, with approximately 12,000 of them being mushrooms [4,5,6]. However, only around 2000 species of mushrooms are edible and/or have medicinal properties [4]. The present study focused on *Agaricus blazei* Murrill (ABM) or Himematsutake (Iwade strain 101), an edible mushroom of the *Agaricaceae* family that was originally identified in Brazil in 1960 [7,8]. Several studies have shown the beneficial effects of ABM in different pathological conditions. For example, ABM derivatives exert an antidiabetic property by accelerating glucose metabolism via the phosphoinositide 3-kinase/Akt pathway, improve hyperlipidemia by activating the peroxisome proliferator–activator receptor-γ pathway or by normalizing the gut microbiome community and protecting against organ injury by inhibiting oxidative stress [9,10,11,12,13]. Anticancer activity is another well-recognized beneficial effect of ABM. Evidence shows that ABM derivatives reduce the proliferation of leukemic cells and cancer cells from the lungs, stomach and pancreas [14,15,16,17]. However, the precise mechanism remains unclear. Previous reports suggested that ABM-derived products can directly inhibit the proliferation or induce the apoptosis of tumor cells or stimulate the killing of malignant cells by host immune cells [7,14,15,16,17,18]. In addition, the inhibitory activity on mutagenesis suggests that ABM might also prevent carcinogenesis [19,20,21].

The most frequent cause of death worldwide is cancer. The World Health Organization reported approximately 10 million cancer-related deaths in 2020 [22]. The leading killer among malignant diseases in 2020 was lung cancer (1,796,144) followed by cancer of the colon and rectum (915,880), liver (830,180), stomach (768,793) and breast (684,996) [22] (https://www.who.int/news-room/fact-sheets/detail/cancer, accessed on 10 June 2021). In Japan, it is estimated that about 70,000 people die from cancer every year. The need for the development of effective therapeutic modalities for cancer is urgent. New anticancer therapeutic modalities, including molecular target therapy and immunotherapy, have been recently developed. Clinical trials with molecules targeting aberrant receptor tyrosine kinase activation or immunotherapy targeting immune checkpoint molecules, including programmed death-ligand 1 (PD-L1) and programmed cell death protein 1 (PD-1), have shown promising results for improving the life expectancy of patients with malignant tumors [23,24,25,26]. PD-L1 from tumor cells inhibits the immune response by binding to PD-1 expressed on the surface of T cells [27]. Inhibitors of PD-L1 and PD-1 restore the cytotoxic activity of T cells against cancer cells by blocking the interaction of both immune checkpoint molecules [27]. However, the development of acquired resistance to these novel anticancer therapies is currently a difficult challenge in clinical practice. Multiple mechanisms of drug resistance have been described in cancer cells [28,29]. Axl, a receptor tyrosine kinase, is an important mediator of drug resistance in cancer [30,31,32]. High tumoral expression of Axl has been associated with enhanced invasiveness, metastatic dissemination, drug resistance and poor prognosis in cancer patients [33,34,35]. Axl is expressed throughout the body by different cells, including immune cells [32]. The activation of Axl may promote tumor invasiveness and drug resistance by inhibiting the host immune response, by inducing the epithelial–mesenchymal transition phenotype or suppressing the apoptosis of malignant cells [34,36,37,38]. Interestingly, a recent study has shown that Axl can stimulate the expression of immune checkpoint molecules and contribute to tumor growth in lung cancer [39].

Subcritical water extraction is a relatively new extraction method of compounds from natural sources that allows one to maintain water in a liquid state by applying high pressure and temperatures between 100 and 374 °C [40]. Water has a high dielectric constant and a high polarity at ambient conditions. A high temperature lowers the dielectric constant, the polarity and weakens the hydrogen bonds of water; thereby water is converted to a less-polar solvent [41]. The solubility, extraction rate and activity of less polar compounds improve when the temperature of the subcritical water is increased [40]. In addition, compared to traditional extraction methods, extraction with subcritical water is more rapid, cost-effective and avoids the use of toxic organic solvents [40].

No study has previously reported whether extracts from the ABM’s fruiting body or mycelium can affect the expression of immune checkpoint molecules or Axl in lung cancer cells. Further, to date, most studies reporting the nutritional or medicinal properties of ABM were conducted using extracts prepared with basic or organic solvents. No study has reported whether subcritical water extracts from ABM also have beneficial properties. In the present study, we hypothesized that subcritical water extracts from ABM inhibit the expression of immune checkpoint molecules and Axl in lung cancer cells.

## 2. Materials and Methods

### 2.1. Preparation of ABM Fruiting Body Extracts at Subcritical Conditions and Increasing Temperatures

ABM culture and extracts were prepared at the Iwade Research Institute of Mycology Corporation using standard protocols. Briefly, the ABM fruiting body was treated with hot water between 80 °C to 100 °C for 3 h to collect the mushroom hot water extraction residue. Part of this residue extract was used to prepare an alkaline extract by treating the residue with 3% to 10% sodium hydroxide solution at temperatures between 25 °C and 40 °C for 24 h. The alkaline extract was then filtered, and the filtrate was sequentially concentrated under reduced pressure before being used in the experiments.

The remaining portion of the ABM hot water extraction residue was used to prepare subcritical water extracts at increasing temperatures. For this purpose, the extraction was performed at megapascal pressures between 2 and 5 and at temperatures of 120 °C, 140 °C, 160 °C, or 200 °C for 5 to 240 min. The collected extract was filtered and lyophilized before using in the experiments.

The polysaccharide FIII-2-b, an ABM fruiting body extract, was produced and provided by the Iwade Research Institute of Mycology Corporation (Tsu, Japan). During the extraction of FIII-2-b, alkalis (sodium hydroxide) and strong acid (ammonium oxalate) are used to facilitate the release of polysaccharides, and sodium borohydride is used to prevent oxidation [41,42]. These solvents pose major hazards during large-scale extraction. In addition, the extraction of FIII-2-b with these solvents is generally performed after hot water extraction followed by successive extraction steps to maximize polysaccharide recovery [41]. Therefore, the lengthy extraction period is another limitation of the alkaline extraction of polysaccharides [41]. In the present study, the extraction rate of FIII-2-b from the fruiting body of ABM was less than 1%. The quality and purity of FIII-2-b polysaccharide were confirmed by comparing the structure with a standard compound using nuclear magnetic resonance spectroscopy as previously described [42].

### 2.2. Preparation of ABM Mycelium Extracts

ABM mycelium (50 g/L) solution was dried in warm air and then used to prepare a subcritical water extract, a hot water extract, and a hot water extract residue subcritical water extract. To prepare the mycelium subcritical water extract, the dried mycelium was placed in a device (Aqua Illusion SG-2, Tohzai Chemical Industry Corporation, Ltd., Osaka, Japan) and subjected to a temperature of 160 °C for 5 min. The mycelium subcritical water extract was then collected, filtered, lyophilized and stored until use. The mycelium hot water extract was prepared by treating the dried ABM mycelium with hot water at temperatures between 90 °C and 95 °C for 3 h. The mycelium hot water extract was then separated, filtered, lyophilized and stored until use. To prepare the subcritical water extract of the mycelium hot water extraction residues, the hot water extract residue was subjected to subcritical water extraction at 160 °C for 5 min. Then, the residue subcritical water extract was collected, filtered, lyophilized and stored until use.

### 2.3. Cell Culture

The human lung adenocarcinoma-derived epithelial cell lines A549 and H3255 were obtained from the American Type Culture Collection (Manassas, VA, USA), the Dulbecco’s Modified Eagle Medium (DMEM) from Sigma-Aldrich (Saint Louis, MO, USA), and the fetal bovine serum (FBS) from Bio Whittaker (Walkersville, MD, USA). L-glutamine, penicillin and streptomycin were from Invitrogen (Carlsbad, CA, USA). The A549 cell line was cultured in DMEM supplemented with 10% fetal bovine serum (Bio Whitttaker, Walkersville, MD, USA), L-glutamine and sodium pyruvate (Nacalai Tesque, Inc., Kyoto, Japan) in a 5% CO_2_/95% air atmosphere at 37 °C.

### 2.4. Cell Stimulation Cell Viability Assay

A549 cells were cultured using 10% FBS DMEM up to sub-confluence and then serum starved overnight before being stimulated with each ABM extract for 24 h at 37 °C and in an atmosphere of 5% CO_2_/95% air before assessing the cells with flow cytometry or PCR. Cell viability was performed using a commercial cell counting kit following the manufacturer’s instructions (CellTiter 96Aqueous, one solution cell proliferation assay, Promega, Madison, WI, USA).

### 2.5. Gene Expression Analysis

We used Sepasol RNA-I Super G reagent (Nacalai Tesque Inc., Kyoto, Japan) to extract the total RNA from A549 cells and then synthesized cDNA from 2 μg of total RNA using oligo-dT primer and Reverse Transcriptase (Toyobo Life Science Department, Osaka, Japan). RT-PCR was then performed using primers described in Table 1. PCR was performed using cycles between 26 to 35, denaturation at 94 °C for 30 s, annealing at 65 °C for 30 s, elongation at 72 °C for 1 min, followed by a further extension at 72 °C for 5 min. The expression of mRNA was normalized against the glyceraldehyde 3-phosphate dehydrogenase (GAPDH) mRNA expression.

### 2.6. Bone Marrow-Derived Dendritic Cells

Eight-week-old female C57BL/6 mice were sacrificed by anesthesia overdose to collect bone marrow cells from the femur and fibula. The cells were washed to remove the debris with RPMI-1640 medium containing 10% heat-inactivated FBS and then cultured for seven days at 37 °C in a humidified 5% CO_2_/95% air atmosphere using RPMI-1640 medium supplemented with 10% heat-inactivated FBS, 2 mM L-glutamine, 100 U/mL penicillin, 100 µg/mL streptomycin, 50 µM µ-mercaptoethanol (Sigma-Aldrich Co, St. Louis, MO, USA) and 200 ng/mL Flt3L (PeproTech, Inc., Rocky Hill, NJ, USA). The harvesting of CD11c + BMDCs was then performed using mouse CD11c microbeads (Miltenyi Biotec GmbH, Bergisch Gladbach, North Rhine-Westphalia, Germany). On day 6 of culture, more than 90% of the cells showed the dendritic cell (CD11c + MHC class II+) phenotype. The dendritic cells (1.2 × 10^6^ cells/mL) were then cultured in the presence of each mycelium extract at a concentration of 20 µg/mL for 48 h before taking samples for analysis. CD86, CD80 and MHC class II expressions on dendritic cells were evaluated using a flow cytometer (FACScan, BD Biosciences, Oxford, UK).

### 2.7. Statistical Analysis

The results are displayed as the mean ± standard deviation of the means (SD) unless otherwise specified. The statistical difference between the two variables was assessed by the unpaired Student’s *t*-test and the difference between three or more variables by one-way analysis of variance using Newman–Keuls’ test for post hoc analysis unless otherwise specified. *p*-value < 0.05 was considered statistically significant. We used GraphPad Prism vs. 7 (GraphPad Software, Inc., San Diego, CA, USA) to perform the statistical analysis.

## 3. Results

### 3.1. Inhibitory Activity of the FIII-2-b Polysaccharide in Lung Cancer Cells 

FIII-2-b is an ABM’s fruiting body extract with reported inhibitory activity against several tumor cells [42]. However, its inhibitory activity has not been reported in lung cancer cells. To validate the inhibitory effect of the ABM’s fruiting body extract on lung cancer, we treated A549 and H3255 lung cancer cell lines with varying concentrations of FIII-2-b and evaluated the number of living cells after 48 h. FIII-2-b significantly inhibited the proliferation of both lung cancer cell lines after 48 h in a dose-dependent manner (Figure 1A). In addition, treatment with the ABM extract for 24 h significantly suppressed the relative mRNA expression of Axl, PD-L1 and PD-L2 in A549 lung cancer cells compared to untreated cells (Figure 1B). These results suggest that FIII-2-b can also suppress cell proliferation and the expression of checkpoint molecules in lung cancer cells.

### 3.2. Subcritical Hot Water Extracts from the ABM’s Fruiting Body Inhibit the Expression of Axl and Immune Checkpoint Molecules

Basic or organic solvents are generally used to extract active products from mushrooms [41]. However, these solvents are hazardous to human health, and thus, the resulting extracts are inappropriate to use as nutritional supplements [41]. To overcome this problem, in the present study, we evaluated whether subcritical water extracts from the ABM’s fruiting body retain the mushroom’s beneficial activity against cancer cells. A549 cells were cultured in the presence of fruiting body samples from ABM prepared by subcritical water extractions at different temperatures for 24 h, and the mRNA expression of Axl, PD-L1 and PD-L2 was evaluated. Cells treated with ABM extracts prepared using an alkaline solvent were the positive controls, and cells treated with the same volume of physiological saline were the negative controls. The relative mRNA expression of Axl, PD-L1 and PD-L2 was significantly reduced in cells cultured in the presence of the fruiting body alkaline extract compared to saline-treated cells. Additionally, all the subcritical water extracts of ABM’s fruiting body prepared at different temperatures significantly decreased the expression of Axl, PD-L1 and PD-L2 in the A549 lung cancer cell line compared to cells treated with saline alone (Figure 2). No significant difference in the inhibitory activity on Axl mRNA expression was observed between the subcritical water extracts at different temperatures. The inhibitory activity on PD-L1 and PD-L2 mRNA expression of the subcritical water extracts at 120 °C was reduced compared to the activity of subcritical extracts at higher temperatures (Figure 2). These findings suggest that subcritical hot water extracts from the ABM’s fruiting body contain products that inhibit the expression of Axl receptor tyrosine kinase and immune checkpoint molecules in lung cancer cells. In addition, these results suggest that subcritical water extracts obtained at a broad range of temperatures (120–200 °C) retain the beneficial properties of the ABM’s fruiting body.

### 3.3. Subcritical Water Extracts from the ABM’s Mycelium Decrease the Expression of Axl, PD-L1 and PD-L2 

The fruiting body rather than the mycelium of ABM is generally used to extract nutritional or medicinal products. However, the natural or artificial cultivation of the ABM’s fruiting body requires adequate environmental conditions (e.g., a stable climate, optimal temperature or humidity) and is at a high risk of contamination. By contrast, mycelium requires less work to cultivate at a large scale, and the culture conditions (e.g., temperature, pH) and contamination are much more easily controllable. However, whether water extracts from ABM’s mycelium have the same beneficial properties as the ABM’s fruiting body is unknown. Here, we treated cultured A549 lung cancer cells with subcritical hot water extract, hot water extract and hot water extraction residue subcritical water extract from the ABM’s mycelium and evaluated the expression of Axl and checkpoint molecules compared to control cells, untreated cells and alkaline extract from ABM’s fruiting body. Cells treated with an ABM’s fruiting body extract prepared using an alkaline solvent were the positive controls, and cells treated with the same volume of physiological saline were the negative controls. Compared to saline-treated cells, the relative expression of Axl, PD-L1 and PD-L2 was significantly decreased in cells treated with the subcritical hot water extract from the ABM’s mycelium (Figure 3). The alkaline extract from the ABM’s fruiting body also significantly inhibited the expression of Axl, PD-L1 and PD-L2 compared to saline-treated cells (Figure 3). The hot water extract and the hot water extraction residue subcritical water extract from the ABM’s mycelium showed no significant effect compared to saline-treated cells (Figure 3). These results suggest that the subcritical hot water extract from ABM’s mycelium has the same properties as the alkaline extract from the ABM’s fruiting body.

### 3.4. ABM’s Mycelium Water Extracts Increase the Expression of Maturation Markers of Dendritic Cells

We next hypothesized that ABM mycelium extracts could enhance the immune response by increasing the expression of maturation markers in dendritic cells. We isolated and cultured in vitro mouse bone marrow-derived cells and differentiated them to dendritic cells by culturing in the presence of Flt3 for seven days. On the 6th day of culture, we added ABM mycelium extracts to the culture and assessed the expression of maturation markers on the 7th day by flow cytometry. Cells treated with an ABM’s fruiting body extract prepared using an alkaline solvent were the positive controls, and cells treated with the same volume of physiological saline were the negative controls. The expression of MHC II, C80 and CD86 was significantly increased in cells treated with subcritical water extract, hot water extract, and hot water extraction residue subcritical water extract from the ABM’s mycelium compared to saline-treated cells (Figure 4A–F). The alkaline extract from the ABM’s fruiting body also significantly enhanced the expression of dendritic cell maturation markers compared to saline controls (Figure 4A–F). These results suggest that ABM’s mycelium can potentiate the immune response by enhancing the activation of dendritic cells.

## 4. Discussion

The present study showed for the first time that water extracts from the fruiting body and mycelium of ABM decrease the surface expression of Axl receptor tyrosine kinase and immune checkpoint molecules and that the water extracts from the ABM mycelium promote maturation of dendritic cells.

The antitumor activity of ABM has been previously documented [7]. In addition, direct tumor cell activity and host immune response-mediated mechanism have been reported [7,42,43,44,45,46]. For example, Matsushita et al. reported that the hot water extract of ABM inhibits cell proliferation by inducing G0/G1 cell cycle arrest and enhances caspase-mediated apoptosis by upregulating the expression of proapoptotic genes in several human pancreatic cancer cell lines [17]. In agreement with these observations, other studies have shown that agaritine, a constituent of ABM, suppresses the proliferation and induces the apoptosis of many leukemic cell lines and that blazein, another ABM derivative, accelerates the apoptosis of lung and gastric cancer cell lines [14,15,16]. On the other hand, Kaneno et al. have shown in a tumor-bearing mouse model that treatment with ABM extracts significantly increases the antitumor activity of natural killer cells and the proliferation and antibody production of lymphocytes [47]. The immune response-stimulating activity has been attributed to the polysaccharide (α(1→4)-glucan-β-(1→6)--glucan–protein complex) component of ABM [46,48]. Furthermore, there is convincing evidence showing that inhibiting the surface expression of Axl and immune checkpoint molecules stimulates the cytotoxic activity of T cells against cancer cells [37,49]. Recent in vitro studies demonstrated that components of some mushrooms (Leucopaxillus giganteus, Pleurocybella porrigens) that do not belong to the Agaricus family could inhibit the expression of Axl, PD-L1 and PD-L2 in cancer cells [50,51]. However, the method for artificially cultivating Leucopaxillus giganteus has not been established, and Pleurocybella porrigens is a poisonous mushroom [50,51]. Therefore, the use of these mushrooms for medicinal or nutritional purposes is not recommendable. In the present study, we showed for the first time that extracts from ABM prepared using only water can reduce the surface expression of Axl and immune checkpoint molecules in lung cancer cells and increase the activation of dendritic cells, suggesting the potential application of ABM water extracts for enhancing immunity against cancer.

Axl belongs to the TAM (Tyro, Axl, Mer) family of transmembrane receptor tyrosine kinases [52]. Axl is activated after binding to its ligands, including the growth arrest-specific 6 and the anticoagulant factor protein S [52]. The ligand-independent activation of Axl may also occur by receptor homodimerization, heteromeric dimerization with a non-TAM receptor, or interaction of the extracellular domains on neighboring cells [53]. Under physiological conditions, the activation of Axl inhibits the immune system to prevent excessive inflammation and autoimmunity and induces the epithelial–mesenchymal transition to promote tissue repair [52]. However, in malignant diseases, this regulatory activity of Axl favors tumor invasion [34,38]. The overexpression of Axl is associated with anticancer therapy drug resistance, invasiveness and poor clinical outcome in various cancers [53]. A small molecule inhibitor of Axl has been demonstrated to suppress tumor growth and metastasis in experimental cancer models [38,53]. The interaction of immune checkpoints, which inhibits T cell activation to prevent excessive inflammation under normal conditions, can also be harnessed by tumor cells to evade the host immune response [27]. The binding of PD-L1 from tumor cells to PD-1 leads to inhibition of cytotoxic T cell activation [27]. Improvement in the clinical outcome of cancer patients treated with PDL-1 or PD-1 inhibitors demonstrates the important role of the immune checkpoints in cancer progression [27]. In the present study, we demonstrated that water extracts from ABM significantly inhibit the expression of Axl, PD-L1 and PD-L2 in lung cancer cells compared to controls. Based on these observations, it is conceivable that the reduction in the expression of Axl, PD-L1 and PD-L2 we found here explains in part the antitumor activity of ABM reported in previous studies.

The use of basic or organic solvents is the standard method to extract compounds from ABM [41]. However, basic or organic solvents are potentially harmful agents, and thus their use for extracting nutritional or medicinal products may be risky [41]. On the other hand, alternative procedures to prepare safe mushroom extracts have been challenging. Here, we prepared extracts from the ABM’s fruiting body using water at high temperatures and applying a subcritical condition. The approach was successful because both the alkaline extract and the subcritical water extract of the ABM’s fruiting body showed similar inhibitory activity on the expression of Axl and immune checkpoint molecules in lung cancer cells. Another challenging work for extracting biologically active compounds from the ABM’s fruiting body is the need for large-scale mushroom cultivation [41]. The growth of ABM’s fruiting body depends on the climate condition, temperature, compost quality and the presence or absence of pathogens’ contamination. In stark contrast, the conditions for the growth of ABM’s mycelium are less strict and more controllable, and contamination with pathogens is less probable, enabling large-scale ABM’s mycelium cultivation to be more feasible. Here, we found that similar to the subcritical water extract from ABM’s fruiting body, the subcritical water extract of the ABM’s mycelium significantly inhibits the expression of Axl and checkpoint molecules. These observations suggest that the water extracts from ABM’s mycelium may be used as an alternative source of inhibitors of Axl and checkpoint molecules in lung cancer cells.

## 5. Conclusions

The present study results showed that the subcritical water extract from ABM’s mycelium significantly decreases the expression of immune checkpoint molecules and Axl receptor tyrosine kinase in lung cancer cells and enhances the expression of surface maturation makers in dendritic cells. These observations suggest that the subcritical water extract from ABM’s mycelium is a promising therapeutic tool that may be used to stimulate the immune response to cancer.

## Figures and Tables

**Figure 1 jof-07-00590-f001:**
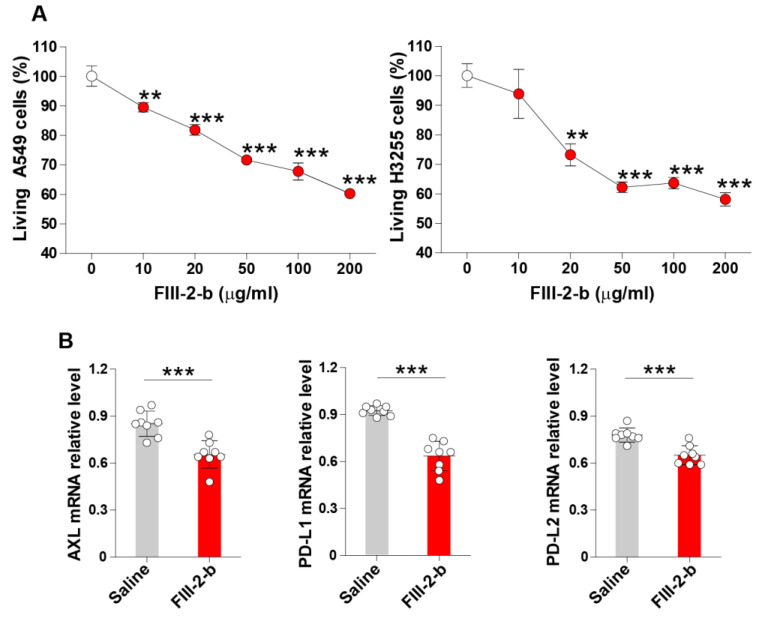
Inhibitory activity of ABM-derived FIII-2-b polysaccharides in lung cancer cells. (**A**), A549, and H3255 lung cancer cells were cultured in the presence of increasing concentrations of FIII-2-b from the ABM fruiting body for 48 h, and then the number of living cells was evaluated by colorimetric assay. The white dots indicate untreated cells and the red dots indicate different concentrations of FIII-2-b in µg/mL. (**B**), A549 cells were treated for 24 h with FIII-2-b (20 µg/mL), and the total RNA was extracted from the cells to evaluate the relative expression of Axl, PD-L1, and PD-1 by RT-PCR. Data are the mean ± S.D. ** *p* < 0.01, *** *p* < 0.001 vs. untreated cells (0 µg/mL) in panel (**A**). *** *p* < 0.001 in panel (**B**).

**Figure 2 jof-07-00590-f002:**
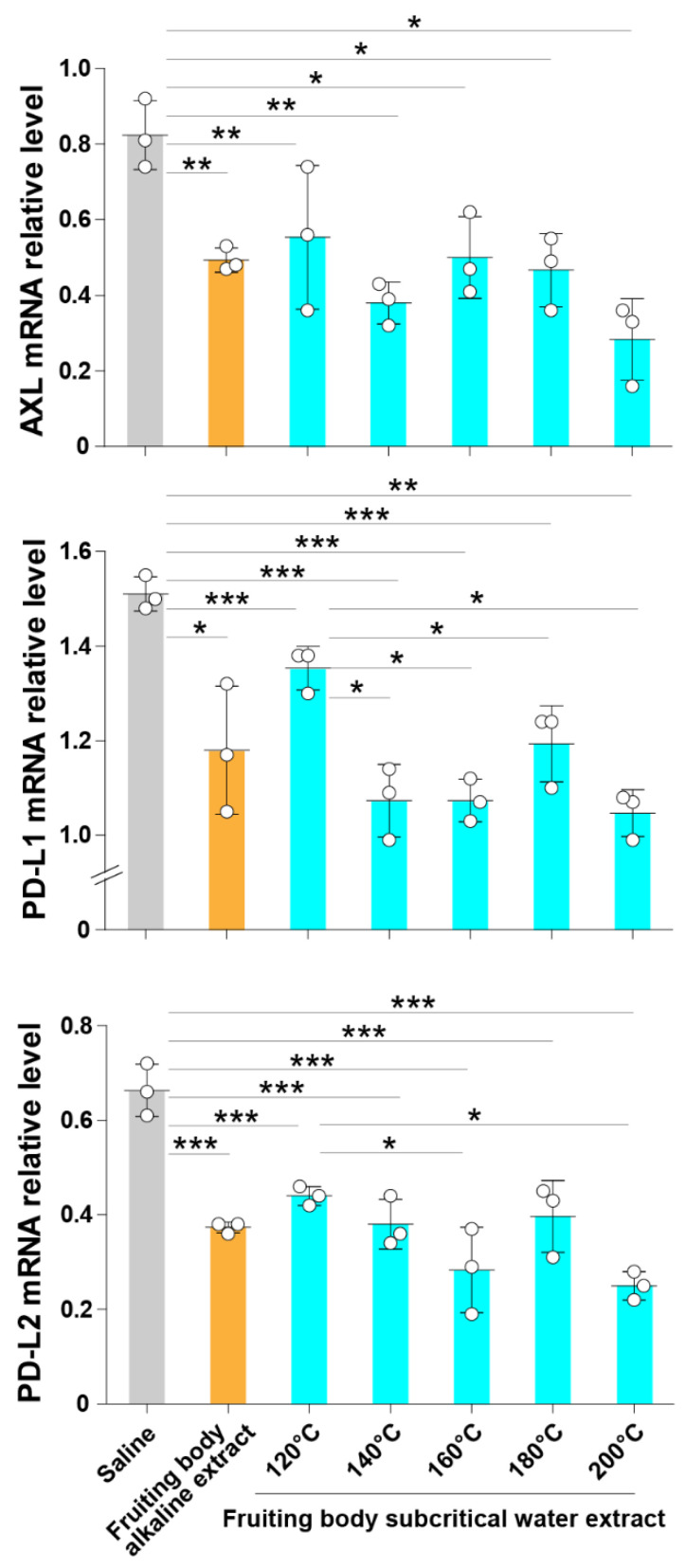
The hot water extracts from the ABM’s fruiting body prepared at subcritical conditions decrease Axl, PD-L1 and PD-L2 expression. A549 cells were cultured in the presence of subcritical water extracts of ABM’s fruiting body prepared at temperatures of 120 °C, 140 °C, 160 °C, 180 °C or 200 °C for 24 h. Cells treated with an alkaline extract from ABM’s fruiting body were used as positive controls, and cells treated with physiological saline alone were used as negative controls. The total RNA was extracted from each treatment group, and the relative expression of Axl, PD-L1 and PD-L2 was evaluated by RT-PCR. Data are the mean ± S.D. * *p* < 0.05, ** *p* < 0.01 and *** *p* < 0.001.

**Figure 3 jof-07-00590-f003:**
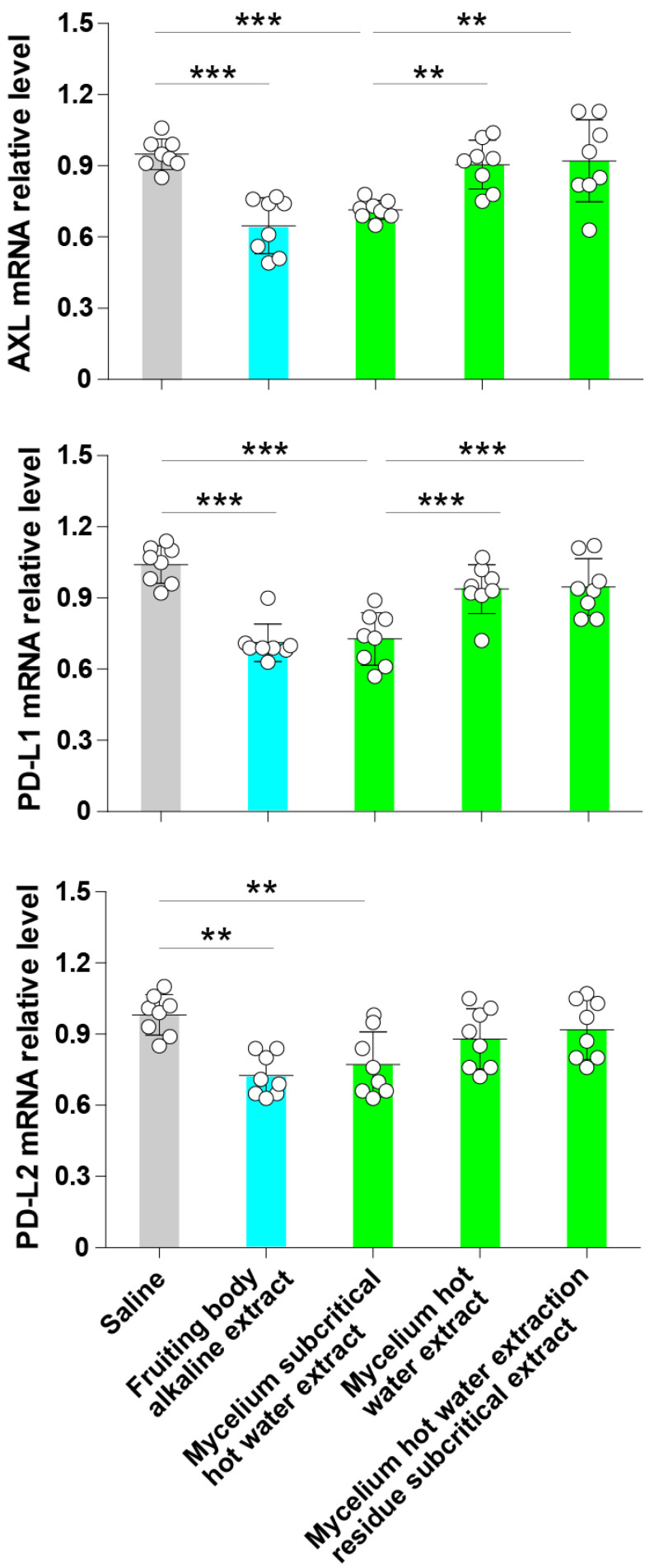
The subcritical water extract from the ABM’s mycelium decreases the expression of Axl, PD-L1 and PD-L2. A549 cells were cultured in the presence of subcritical water extract, hot water extract, and subcritical water extraction residue extracts from ABM’s mycelium for 24 h. Cells treated with an alkaline extract from ABM’s fruiting body were used as positive controls, and cells treated with physiological saline alone were used as negative controls. The total RNA was extracted from each treatment group, and the relative expression of Axl, PD-L1 and PD-L2 was evaluated. Data are the mean ± S.D. The white dots represent the levels of the measurements in each group. ** *p* < 0.01 and *** *p* < 0.001.

**Figure 4 jof-07-00590-f004:**
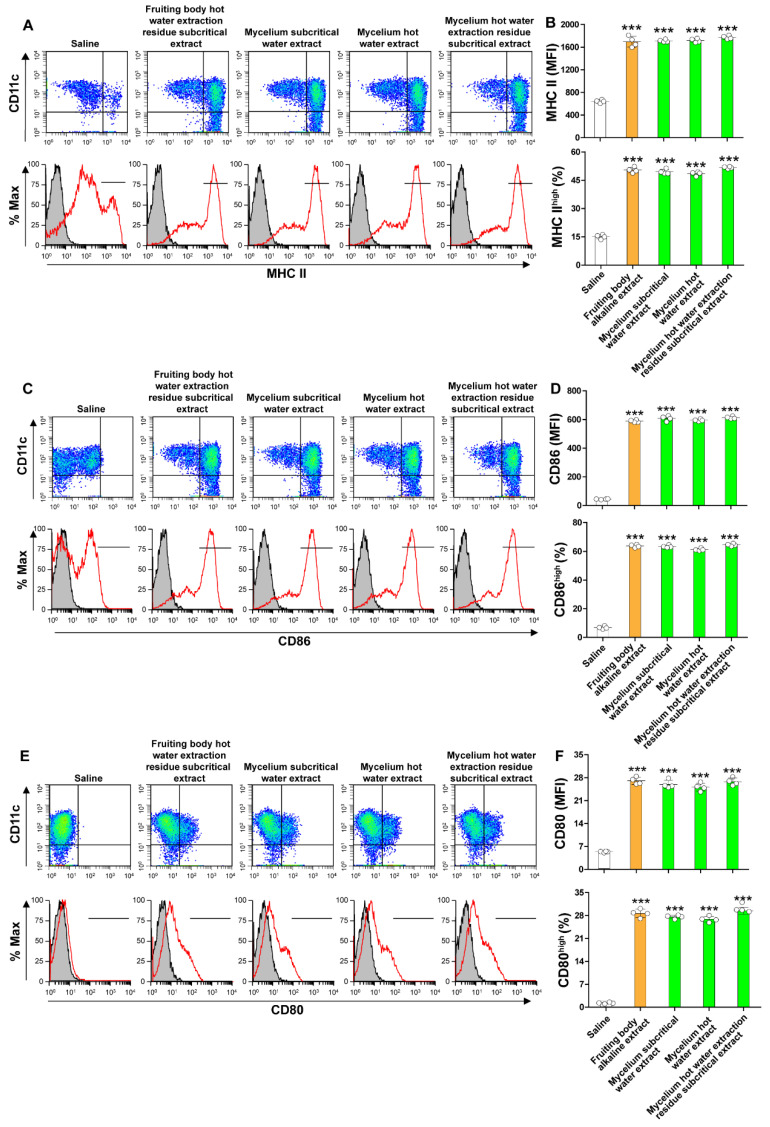
The ABM’s mycelium extracts increase the expression of maturation markers of dendritic cells. Bone marrow-derived dendritic cells were cultured in the presence of alkaline extract, hot water extract, and subcritical hot water residue extracts from ABM mycelium for 24 h. Cells treated with an alkaline extract from ABM’s fruiting body were used as positive controls, and cells treated with physiological saline alone were used as negative controls. The surface expression of MHC II (**A**,**B**), CD86 (**C**,**D**), CD80 (**E**,**F**) was evaluated by flow cytometry. Data are the mean ± S.D. *** *p* < 0.001 vs. saline-treated cells. The gray areas indicate the isotype antibody control. The red lines indicate the specific antibodies against MHC II, CD86 or CD80. MFI is the abbreviation of mean fluorescence intensity.

**Table 1 jof-07-00590-t001:** PCR primers.

Gene	Direction	Sequence (5′ > 3′)	Length	Tm (°C)	Ref	Position	Product
GAPDH	Forward	GGAGCGAGATCCCTCCAAAAT	21	61.6	NM_001256799	108–128	197 bp
	Reverse	GGCTGTTGTCATACTTCTCATGG	23	60.9		304–282	
AXL	Forward	TGCCATTGAGAGTCTAGCTGAC	22	63.4	NM_001699	2311–2322	218 bp
	Reverse	TTAGCTCCCAGCACCGCGAC	20	71.7		2528–2509	
PD-L1	Forward	GGACAAGCAGTGACCATCAAG	21	60.9	NM_014143	500–520	235 bp
	Reverse	CCCAGAATTACCAAGTGAGTCCT	23	61.3		734–712	
PD-L2	Forward	ACCGTGAAAGAGCCACTTTG	20	60.5	NM_025239	206–225	121 bp
	Reverse	GCGACCCCATAGATGATTATGC	22	60.7		326–305	

## Data Availability

All data are available from the corresponding author under reasonable request.

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
