# Peer review of "Subcritical Water Extracts from *Agaricus blazei* Murrill’s Mycelium Inhibit the Expression of Immune Checkpoint Molecules and Axl Receptor"

_jof, 2021, doi:10.3390/jof7080590_

Round 1
Reviewer 1 Report
Authors attempted to describe Subcritical Water Extracts from Agaricus blazei Murrill's Mycelium Inhibits the Expression of Immune Checkpoint Molecules and Axl receptor. MS may be considered for publication after receiving proper responses for the following points:
1. Authors described the extraction method of FIII-2-b polysaccharide in the MS, but didn’t mention the extraction rate of FIII-2-b polysaccharide. How to confirm the quality of FIII-2-b polysaccharide?
2. In this MS, the author showed the inhibitory activity of FIII-2-b polysaccharides, but didn’t mention the cytotoxic activity. If this information can be provided, it will be more safe to use FIII-2-b polysaccharides.
Author Response
Response to questions of Reviewer 1
Comment 1
Authors attempted to describe Subcritical Water Extracts from Agaricus blazei Murrill's Mycelium Inhibits the Expression of Immune Checkpoint Molecules and Axl receptor. MS may be considered for publication after receiving proper responses for the following points:
1. Authors described the extraction method of FIII-2-b polysaccharide in the MS, but didn’t mention the extraction rate of FIII-2-b polysaccharide. How to confirm the quality of FIII-2-b polysaccharide?
Response
First of all, we are very thankful to the Reviewer for his/her comments that has substantially improved the manuscript.
As suggested we have described the extraction rate of FIII-2-b and provided information on how the quality of the product was confirmed.
Please see page 3, lines 111 to 121 in the revised version of the manuscript.
The text is also described below for your convenience:
“During the extraction of FIII-2-b, alkalis (sodium hydroxide) and strong acid (ammonium oxalate) are used to facilitate the release of polysaccharides, and sodium borohydride is used to prevent oxidation [41, 42]. These solvents pose major hazards during large-scale extraction. In addition, extraction of FIII-2-b with these solvents are generally performed after hot water extraction followed by successive extraction steps to maximize polysaccharide recovery [41]. Therefore, the lengthy extraction period is another limitation of alkaline extraction of polysaccharides [41]. In the present study, the extraction rate of FIII-2-b from the fruiting body of ABM was less than 1%. The quality and purity of FIII-2-b poly-saccharide were confirmed comparing the structure with a standard compound using nuclear magnetic resonance spectroscopy as previously described [42].”
Comment 2
2. In this MS, the author showed the inhibitory activity of FIII-2-b polysaccharides, but didn’t mention the cytotoxic activity. If this information can be provided, it will be more safe to use FIII-2-b polysaccharides.
Response
In the present study, we have not studies the cytotoxicity of the FIII-2-b because we have not done experiments in vivo. However, previous studies describing the anti-cancer activity of FIII-2-b using in vivo experimental models showed no clear data regarding the cytotoxicity or adverse effects during the treatment with FIII-2-b. We are planning to perform studies on the safety of these products in the future.

Reviewer 2 Report
The study shows that different types of extracts from Agaricus blazei Murill (ABM) inhibit the expression of immune checkpoint molecules and the Axl receptor in human lung cancer cells , and also activate mouse bone marrow derived dendritic cells. The effect of subcritical water extracts from both the fruiting body and the mycelium of ABM on these parameters were comparable to that of alkaline extracts from the fruiting body. In the practical world the production of extracts from the mycelium of ABM is simpler and more suited for a large scale setting than production of extracts from the fruiting body. The study shows that subcritical water extracts of the mycelium of ABM, i.e. extracts prepared without the use of potentially harmful basic or organic solvents, may have a potential in the treatment of malignant diseases. The study is therefore of considerable interest.
The concept is well founded and the conduct of the study is scientifically adequate.
There are some minor remarks:
- Introduction: The procedure of obtaining subcritical water extracts in general should be explained.
- Materials and Methods 2.1.
In last sentence the expression " ...many difficult and expensive steps (are required to prepare the FIII-2b polysaccharide)" is not very precise and should be replaced.
Results. 3.1 "FIII-2-b is an ABM fruiting body extract with reported effect against several tumor cells. However, it's inhibitory activity has been reported in lung cancer". The word however does not seem suitable. Perhaps it maybe replaced with "In particular(?).
Figure 1A. The units of the x- axis should be defined. The significance of the asterisks (∗) over the curves should be explained. Figure 1B. The significance of three asterisks (∗∗∗) in the figure should be explained.
3.2 ( Subcritical hot water extracts....) A possible effect (or no effect) of the temperature used in producing the subcritical hot water extracts on the effect of these extracts on the decrease of Axl, PD-L1,PD-L2 should be commented (for ex. 120°C vs. 200°C).
Figure 3. The significance of three asterisks (∗∗∗) on the two upper figures should be explained
Author Response
Response to questions of Reviewer 2
Comment 1
The study shows that different types of extracts from Agaricus blazei Murill (ABM) inhibit the expression of immune checkpoint molecules and the Axl receptor in human lung cancer cells , and also activate mouse bone marrow derived dendritic cells. The effect of subcritical water extracts from both the fruiting body and the mycelium of ABM on these parameters were comparable to that of alkaline extracts from the fruiting body. In the practical world the production of extracts from the mycelium of ABM is simpler and more suited for a large scale setting than production of extracts from the fruiting body. The study shows that subcritical water extracts of the mycelium of ABM, i.e. extracts prepared without the use of potentially harmful basic or organic solvents, may have a potential in the treatment of malignant diseases. The study is therefore of considerable interest.
The concept is well founded and the conduct of the study is scientifically adequate.
Response
We are very grateful to the positive comment of the Reviewer and for his/her important suggestions that substantially improved the quality of the manuscript.
Comment 2
Introduction: The procedure of obtaining subcritical water extracts in general should be explained.
Response
As suggested by the reviewer we have added a paragraph in the Introduction section explaining the procedure of subcritical water extraction.
Please see page 2, lines 79 to 87 in the revised manuscript.
The text is also described below for your convenience:
“Subcritical water extraction is a relatively new extraction method of compounds from natural sources that allows to maintain water in a liquid state by applying high pressure and temperatures between 100 and 374 ºC [40]. Water has a high dielectric constant and a high polarity at ambient conditions. A high temperature lowers the dielectric constant, the polarity and weakens the hydrogen bonds of water, thereby water is converted to a less-polar solvent [41]. The solubility, extraction rate and activity of less polar compounds improve when the temperature of the subcritical water is increased [40]. In addition, compared to traditional extraction methods, extraction with subcritical water is more rapid, cost-effective and avoids the use of toxic organic solvents [40].”
Comment 2
Materials and Methods 2.1.
In last sentence the expression " ...many difficult and expensive steps (are required to prepare the FIII-2b polysaccharide)" is not very precise and should be replaced.
Response
As suggested by the Reviewer we have replaced that sentence and added a more detailed explanation.
Please see page 3, lines 111 to 121 in the revised manuscript.
The text is also described below for your convenience.
“During the extraction of FIII-2-b, alkalis (sodium hydroxide) and strong acid (ammonium oxalate) are used to facilitate the release of polysaccharides, and sodium borohydride is used to prevent oxidation [41, 42]. These solvents pose major hazards during large-scale extraction. In addition, extraction of FIII-2-b with these solvents are generally performed after hot water extraction followed by successive extraction steps to maximize polysaccharide recovery [41]. Therefore, the lengthy extraction period is another limitation of alkaline extraction of polysaccharides [41]. In the present study, the extraction rate of FIII-2-b from the fruiting body of ABM was less than 1%. The quality and purity of FIII-2-b poly-saccharide were confirmed comparing the structure with a standard compound using nuclear magnetic resonance spectroscopy as previously described [42].”
Comment 3
Results. 3.1 : "FIII-2-b is an ABM fruiting body extract with reported effect against several tumor cells. However, it's inhibitory activity has been reported in lung cancer". The word however does not seem suitable. Perhaps it maybe replaced with "In particular (?).
Response
We thank the Reviewer for pointing out our error. Actually, the sentence should have been a negative sentence, and thus why we used the word “However”.
In the revised version we have corrected the expression.
Please see page 5, lines 184 to 186 in the revised manuscript.
The text is also described below:
“FIII-2-b is an ABM's fruiting body extract with reported inhibitory activity against several tumor cells [42]. However, its inhibitory activity has not been reported in lung cancer cells.”
Comment 4
Figure 1A. The units of the x- axis should be defined. The significance of the asterisks (∗) over the curves should be explained. Figure 1B. The significance of three asterisks (∗∗∗) in the figure should be explained.
Response
As suggested we have added the label and unit of the x axis to Figure 1A and added explanation about the (*). The significance of (***) was also explained.
Please see pages 5 and 6, lines 194 to 213, Figure 1 in the revised manuscript.
Comment 5
3.2 ( Subcritical hot water extracts....) A possible effect (or no effect) of the temperature used in producing the subcritical hot water extracts on the effect of these extracts on the decrease of Axl, PD-L1,PD-L2 should be commented (for ex. 120°C vs.200°C).
Response
We have analyzed the effects of the different temperatures and explained the results.
Please see page 6, lines 230 to 234.
The text is described below:
“No significant different in the inhibitory activity on Axl mRNA expression was observed between the subcritical water extracts at different temperatures. The inhibitory activity on PD-L1 and PD-L2 mRNA expression of the subcritical water extracts at 120ºC was reduced compared to the activity of subcritical extracts at higher temperatures (Figure 2).”
Comment 6
Figure 3. The significance of three asterisks (∗∗∗) on the two upper figures should be explained.
Response
We have added the explanations as suggested by the Reviewer.
Please see page 8, lines 294 to 325, and the Figure 3.
